# Painting with odors: How olfactory stimuli influence artistic expression, emotional response, visual perception, and object selection

Zahra Davoudi[1,2], Nobuyuki Sakai[1,2,3]*

1 Vision to Connect, COI-NEXT, Tohoku University, Sendai, Japan, 2 Department of Psychology, Graduate School of Arts and Letters, Tohoku University, Sendai, Japan, 3 Advanced Institute of So-Go-Chi Informatics, Tohoku University, Sendai, Japan

* nob_sakai@tohoku.ac.jp

## Abstract

In recent years, the relationship between olfaction and vision has received increasing attention. We combined psychology and art to investigate how two specific odors, strawberry and rose, influence visual perception, artistic expression, creativity, and object selection across two studies. In Study 1, 24 participants created paintings inspired by rose and strawberry odors. Sixty additional participants evaluated these paintings using seven semantic differential scales. In the final phase, another 60 participants rated the odors using the same scales. In Study 2, 60 participants were randomly assigned to one of three rooms: a strawberry-odorized room, a rose-odorized room, or a control room with no odor. Five artificial objects (strawberries, lemons, and roses) were placed on a table in each room. Without being told about the presence of odors, the participants were asked to select one object and paint it. Afterward, they reported the odor they perceived and selected the associated colors using a standardized color panel. This design aimed to determine whether specific odors influenced object selection and visual perception. Together, these experiments provide converging evidence suggesting consistent odor-color associations and indicate that odors may influence visual perception and object selection.

## Introduction

The attributes of odor stimuli contribute to the cross-modal correspondence between olfactory and visual characteristics, such as color dimensions [1]. Recent studies indicate that ambient olfactory cues can bias rapid visual attention. For example, in an involuntary "attentional capture" paradigm, Michael et al. found that background odors modulated reflexive orienting to sudden luminance onsets [2]. In a follow-up task, the same group showed that a trigeminal (irritant) odor (allyl isothiocyanate) increased both the magnitude and duration of attentional capture. In contrast, a pleasant rose-like odor (phenylethyl alcohol) effectively abolished the capture effect

**Data availability statement:** All raw data and analysis scripts underlying the findings of this study are available from the Open Science Framework (OSF) at DOI https://doi.org/10.17605/OSF.IO/P5SBH.

**Funding:** This study was supported by the Japan Science and Technology Agency (JST), Grant Number JPMJPF2201 (NS). The funders had no role in study design, data collection and analysis, decision to publish, or preparation of the manuscript.

**Competing interests:** The authors have declared that no competing interests exist.

and generally slowed processing [3]. Parallel findings have emerged in visual search and scanning tasks. Seo et al. used eye tracking to show that smelling a congruent odor caused observers to make increasingly longer fixations on matching objects than in an odorless condition [4], while Seigneuric et al. similarly found that congruent ambient scents led participants to locate odor-associated targets more quickly during free viewing [5]. Chen et al. extended this by demonstrating, in dot-probe and search approaches, that a smell reflexively draws the attentional "spotlight" to a semantically matching image, facilitating detection even against competing distractors [6]. Neurophysiological evidence also supports odor-driven attentional shifts. Zhang et al. recorded electroencephalogram activities during an Attention Network Test and found that an unpleasant odor elicited larger early N1/N2 components to alerting cues and faster behavioral alerting responses, suggesting heightened preparatory attention to negative odors [7]. Castellotti et al. reported that ambient fruit odors congruent with the search target improved visual search performance, yielding higher accuracy and faster responses for matching targets than for incongruent or no-odor trials [8]. These behavioral and neurocognitive studies suggest that background odors can bias rapid visual attentional deployment via affective congruency and arousal mechanisms, thereby altering eye movements, reaction times, and early neural processing of visual cues. Odor-color correspondence should be interpreted within the broader framework of odor-association learning, rather than isolated cross-modal mapping. Evidence from taste-odor learning demonstrates that odors are preferentially associated with the emotional (hedonic) aspects of gustatory information rather than with qualitative sensory features. Using higher-order conditioning paradigms, Onuma and Sakai demonstrated that odors acquire associative value primarily through emotional valence, with second-order conditioning emerging only when affective information is available. In contrast, the associations based on sensory quality alone were comparatively weak [9]. Complementary human studies have further indicated that visual information systematically modulates odor perception via top-down cognitive processes. Visual cues such as colors or images evoke learned expectations that influence perceived odor intensity, preference, and identification [10]. These findings suggest that odor-visual correspondences, including odor-color associations, arise from learned multimodal associations grounded in hedonic evaluations and cognitive expectations. From this perspective, color functions as a visual manifestation of affective and semantic associations linked to odors, rather than as a direct sensory counterpart, providing a theoretical basis for understanding the odor-induced modulation of visual expression. When an individual smells an odor, the associated color can significantly influence olfactory perception and source identification [11,12]. Pleasant odors tend to be associated with warmer, lighter, and more saturated colors, whereas unpleasant odors are often associated with darker or duller hues [13]. Supporting this association, Levitan et al.[14] demonstrated that odors evoke emotional responses that subsequently influence perceptual judgments, suggesting that odor-color associations emerge from an interaction between learned semantic links and affective processing. For example, lemon odor was found to be reliably associated with yellow, reflecting both learned associations with fruits and perceptual impressions of

freshness and brightness [15,16]. One effective way to investigate how people perceive odors is through painting [16,17]. People can visually represent the colors and shapes they associate with the odors they experience. Relevant research [15] has shown that paintings enhance emotional awareness.

In this study, paintings were used to visualize the perception of odors. In the first study, painters were exposed to specific odors, strawberry and rose, in a controlled environment and were asked to create paintings reflecting their sensory and emotional responses. Participants freely expressed their perceptions, allowing the unconscious influence of these odors to manifest visually. Strawberry and rose odors were selected because they provide a theoretically controlled contrast along the semantic and ecological dimensions, while remaining comparable in hedonic valence. Both odors are familiar and readily identifiable across cultures, making them suitable for use with international student populations [14,18]. Although strawberry and rose odors are both typically perceived as pleasant and are often associated with red or pink hues, converging evidence indicates that odor-color associations are not determined by hedonic valence alone, but are strongly influenced by semantic category and object-based knowledge [15,19]. Strawberry has a sweet, fruity, and appetizing odor that is closely linked to gustatory experiences and food-related representations, whereas rose has a floral odor associated with botanical environments and culturally embedded symbolic meanings, such as elegance or romance [20]. In the second study, participants were placed in rooms with either a strawberry or rose odor or in a control room with no odor and asked to select one object from a set of five before painting it. They were unaware of the presence of odors, allowing their choices to reflect unconscious influences.

Although previous studies [10,11,21] have examined how visual cues influence olfactory perception, few [1,4] have explored how odors affect visual perception. To address this gap, this study aimed to investigate whether odors can influence visual perception and artistic expression through paintings.

This study had three main goals. The first goal was to visualize odor perception through painting and examine whether individuals perceived similar ranges of colors from the same odor. The second goal was to illustrate how odors influence people's choices. The third goal was to highlight the importance of olfaction in color selection and evaluation. Using these two experimental studies, we aimed to deepen understanding of how olfactory stimuli interact with visual and emotional processes.

## Study 1

This study investigated the influence of olfactory stimuli on visual perception, artistic expression, perceptual evaluation, and emotional response. Specifically, it explored how exposure to distinct odors, strawberry and rose, affects participants' use of colors in paintings, the evaluation of these paintings by others, and the perception of the odors themselves. By examining these multisensory interactions, this study aimed to deepen understanding of the interplay between the olfactory and visual modalities in creative processes and emotional judgments. Novak et al.[22] found that olfactory-visual integration facilitates the perception of subthreshold negative emotions. Their study revealed that even subtle olfactory cues could influence the perception of negative emotions in visual stimuli, highlighting the sensitivity of the olfactory system to emotional contexts. This underscores the role of olfactory cues in enhancing emotional perception, even when visual cues alone do not convey strong emotional content.

It was hypothesized that exposure to strawberry and rose odors would lead participants to use distinct color palettes in their paintings and that paintings inspired by these odors would be evaluated differently.

### Materials and methods

**Participants.** Twenty-four university students (11 women, 13 men; mean age = 22.4, standard deviation [SD] = 2.1) were recruited from Tohoku University, Sendai City, Japan, to participate in the first painting phase. In the second phase (evaluation), 60 participants (30 women and 30 men; mean age = 23.5, SD = 1.8) evaluated the paintings. In the final rating phase, an additional 60 participants (30 women and 30 men; mean age = 23.5, SD = 1.8) rated the odors. All the

participants reported normal vision and no olfactory impairment. This study was conducted from January 1, 2024 to March 31, 2024, at the Kawauchi Minami Campus, Tohoku University. Written informed consent was obtained from all participants, and the study was approved by the Ethical Committee for Medical Research on Humans at the Kawauchi Minami Campus, Tohoku University (Approval No. 2023−011).

**Materials.** During the painting phase, an Aroma Breeze Nova Diffuser (Patchoul, Japan) was used to consistently deliver olfactory stimuli. This diffuser used an internal absorbent cotton pad as the odor carrier, and the essential oil was applied directly to the cotton, which served as the sole source of odor delivery. Strawberry (Ishida Foods Corporation, Tokyo, Japan) and rose essential oils (Tomizawa, Tokyo, Japan) were presented via the diffuser for a brief direct inhalation before the painting session. A 24-color acrylic paint set spanning the full spectrum of primary, secondary, and neutral tones (Fantastory, Model Number: FAN-BOT-60X24-XBZ-EU), 30 brushes of varying sizes, and A4 canvases were used for standardized artistic production. All paintings were photographed with a digital camera under controlled indoor lighting conditions. The camera parameters (ISO, aperture, and white balance), camera-painting distance, and background were kept constant across all recording sessions. In the evaluation phase, the participants evaluated images of paintings presented via Google Forms on a PC (MacBook Pro 13-inch 2020, Apple Inc., USA). In the rating phase, the same descriptive word pairs were used for consistency and control during the assessments. The participants were asked to rate the odors using descriptors presented on the screen (MacBook Pro) with PsychoPy (v2024.1.3; Jonathan Peirce and colleagues). Additionally, the LightColor 3D Munsell Color Panel (Nihon Shikiken, Tokyo, Japan) was used to select matching colors. This panel, designed for professional color evaluation, included 298 colors across 10 brightness levels with oil color labels and acrylic laminated charts in the A5 format.

**Procedure and design.** The odor stimuli were delivered as vaporized aromas using the Aroma Breeze Nova Diffuser, in which the essential oil was applied to an internal absorbent cotton pad that served as an odor source. For each experimental session, a fixed volume of undiluted essential oil (10 drops, approximately 0.5 mL) was applied to the cotton to ensure consistency across trials. In Study 1, odor exposure was delivered as a brief direct inhalation rather than as continuous room odorization. The participants were seated approximately 20–30 cm from the diffuser and instructed to inhale the odor for approximately 10 s before beginning the painting task. The diffuser was not used to maintain a continuous ambient odor during painting. After initial exposure, painting was performed without additional direct inhalation. Airborne odor concentrations and intensities were not chemically calibrated. During the painting phase, each painter was asked to smell the strawberry and rose odors on different days and paint what they perceived. A minimum interval of 48 hours was maintained between sessions to minimize potential carryover effects. Before the start of each session, the experimenter confirmed that no residual odor from the previous condition remained in the experimental room. After completing each painting, they were asked to smell the odor again and report the colors they imagined from the odor and its name. To avoid cross-participant odor contamination, the room was thoroughly ventilated for 10 min between sessions. Windows were opened and an electric fan was used to dissipate any remaining odor, ensuring that each participant started in a neutral olfactory environment. The order of the odor conditions in Study 1 was counterbalanced across participants. Each painter experienced two odor conditions (strawberry and rose) in a random order. In the second phase, the participants (evaluators) were asked to evaluate the paintings using eight pairs of words on a 7-point Likert scale. A semantic differential scale [23] was used to evaluate the representation of the paintings. In the third rating phase, the participants (raters) were asked to rate each odor independently using the same semantic differential scale as in the evaluation phase. After rating, the participants were asked to name the odor. Fig 1 shows the experimental setups for each phase.

**Analysis.** Python and R were used for the statistical analyses. Image preprocessing and color extraction were performed using Python software. Images were loaded using the Python Imaging Library (PIL) and converted to an 8-bit RGB color space using the convert ("RGB") function, and resized to a standardized resolution of 200 × 200 pixels using PIL's default resampling parameters to ensure uniform spatial sampling across images.

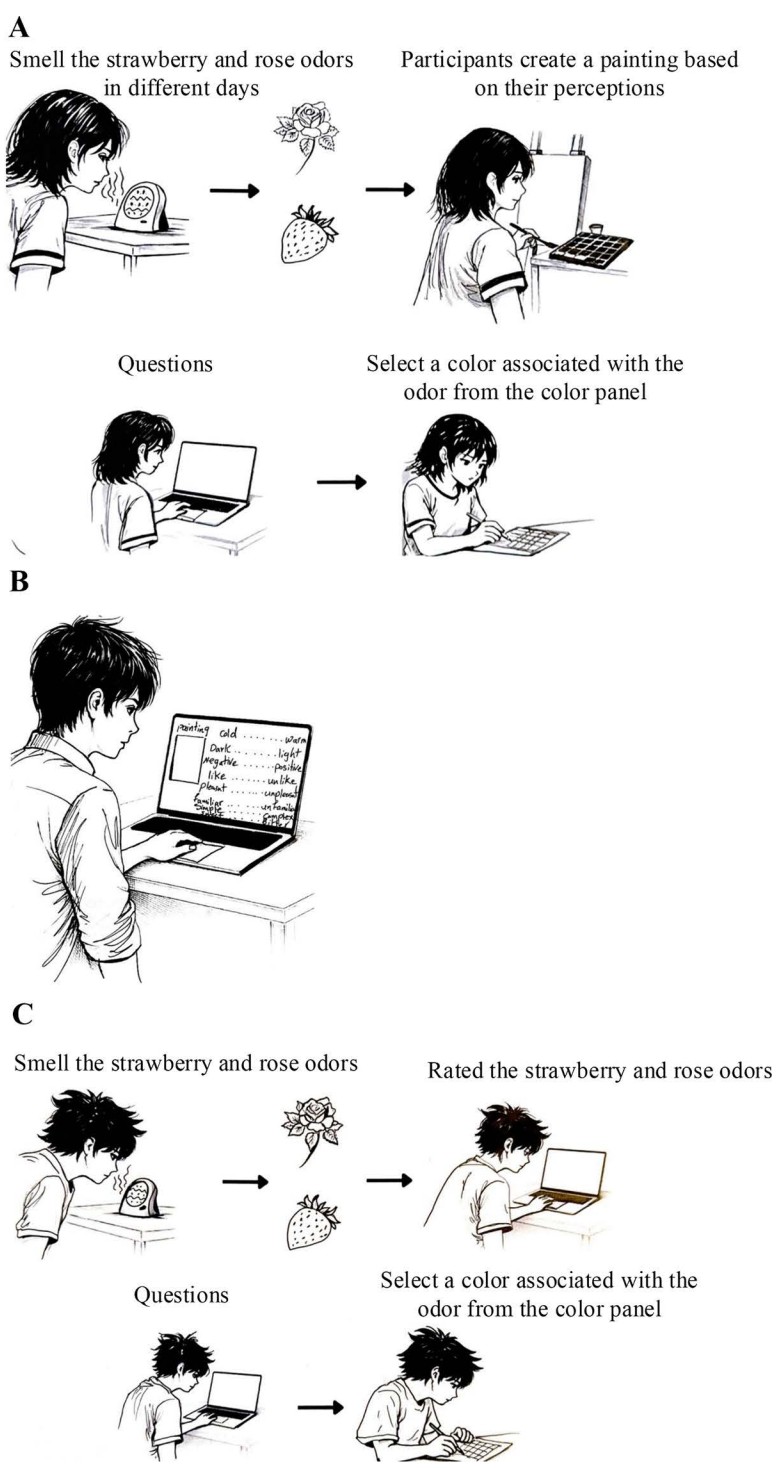

**A**

Smell the strawberry and rose odors
in different days

Participants create a painting based
on their perceptions

Questions

Select a color associated with the
odor from the color panel

**B**

**C**

Smell the strawberry and rose odors

Rated the strawberry and rose odors

Questions

Select a color associated with the
odor from the color panel

**Fig 1. Experimental setup across the three phases of Study 1.** (A) First phase. (B) Second phase. (C) Third phase.

Pixel values were reshaped into RGB triplets and subjected to k-means clustering (k = 25, random_state = 0, n_init = 10) to estimate the dominant color centroids for each painting. Images were acquired under controlled indoor lighting conditions with fixed camera parameters (ISO, aperture, and white balance) as well as constant camera-to-painting distance and background. No additional histogram equalization, channel scaling, intensity normalization, or post-hoc white-balance correction was applied because the acquisition conditions were standardized to minimize illumination variability while preserving the original pixel-value distributions for clustering.

The number of clusters was fixed at 25 for all the paintings, corresponding to a 5 × 5 color palette. This selection reflects a balance between capturing fine-grained color variations and maintaining comparability across images, which is consistent with the established approaches in digital color analysis and computational aesthetics [24].

Exploratory factor analysis (EFA) was conducted on the semantic differential ratings to identify underlying perceptual dimensions. Prior to factor extraction, sampling adequacy was supported using the Kaiser-Meyer-Olkin (KMO) measure and Bartlett's test of sphericity. Factors were extracted using principal axis factoring (PAF) with varimax rotation. The number of factors was determined based on the eigenvalues (>1), scree plot inspection, and parallel analysis. Factor scores were estimated using the Ten-Berge method and were used in subsequent statistical analyses. We analyzed the data using linear mixed-effects models (LMMs) with odor as a fixed effect and random intercepts for the painters and evaluators to account for repeated measurements. The models were fitted using maximum likelihood estimation in the lmerTest package, and the degrees of freedom were approximated using the Satterthwaite method. Planned pairwise comparisons between odor conditions were conducted using the estimated marginal means (EMMs) with Bonferroni adjustments across outcomes. Effect sizes are reported as partial $\eta^2$ for fixed effects and Cohen's $d$ for pairwise contrasts. Data visualizations, such as color distribution graphs, were created in R using packages such as ggplot2 to effectively represent the patterns and relationships in the data. The Munsell color values were converted to RGB codes using the online tool provided by QConv (https://qconv.com/en/convert-munsell-to-rgb), which facilitates accurate color representation in the data.

**Results.** All paintings are shown in Fig 2, which shows the dominant colors extracted from each painting. The results indicated that paintings produced under the same odor conditions exhibited similar color-use patterns. The participants tended to use warmer hues such as yellow, pink, and orange for the strawberry condition, and cooler hues such as blue and green for the rose condition.

Each row represents an individual participant's artwork for both odors.

Descriptions of the odors by the participants are shown in Table 1. For the strawberry odor, the participants frequently labeled the strawberry odor as "strawberry" (37.50%), "gummy" (37.50%), and "fruit" (20.83%), with a smaller percentage associating it with "candy" (4.17%). For the rose odor, the participants predominantly labeled it as "rose" (50.00%) and "flower" (41.67%), with smaller associations with "detergent" (4.17%) and "shampoo" (4.17%).

EFA of the semantic rating scales revealed a two-factor solution. Based on eigenvalues greater than 1 ($\lambda_1 = 4.32$; $\lambda_2 = 1.07$), the two factors together explained 67.3% of the total variance, with Factor 1 accounting for 38.5% and Factor 2 for 28.8% of the variance. Factor 1 showed high loadings (|loading| ≥ 0.40) on the cold-warm, negative-positive, and dark-light scales, reflecting a broad affective evaluative dimension. Factor 2 exhibited high loadings on the pleasant-unpleasant, like-unlike, simple-complex, familiar-unfamiliar, and sweet-bitter scales, capturing perceptual valence and complexity dimensions. Based on their loading patterns, Factor 1 was labeled Affective Dimensionality and Factor 2 was labeled Perceptual Valence and Complexity.

Paintings created under the strawberry odor condition were evaluated as lighter, warmer, and more positive than those under the rose odor condition. However, preference evaluations were the opposite; evaluations of pictures associated with the rose odor were higher than those associated with the strawberry odor (Table 2 and S1 Table). The Affective Dimensionality scores (Factor 1) were higher for the strawberry odor, indicating that the participants perceived these paintings

**Fig 2. Paintings produced by participants (strawberry and rose odors), alongside corresponding color charts.**

as more emotionally intense or vivid. However, the rose odor received higher scores on Factor 2, suggesting that it was perceived as less pleasant, less liked, and more unfamiliar (Fig 3).

The effects of odor on painting evaluations were examined using linear mixed-effects models, with odor specified as a fixed effect and random intercepts for the painters and evaluators, thereby accounting for repeated paintings by the same painter and ratings by the same evaluator.

**Table 1. Participants' labels (strawberry and rose odors) associated with their paintings during the painting phase.**

| Word | Strawberry odor (%) | Rose odor (%) |
|---|---|---|
| Candy | 4.17 | |
| Detergent | | 4.17 |
| Flower | | 41.67 |
| Fruit | 20.83 | |
| Gummy | 37.50 | |
| Rose | | 50.00 |
| Shampoo | | 4.17 |
| Strawberry | 37.50 | |

**Table 2. The rotated factor matrix derived from factor analysis.**

| Variables | Factor 1 | Factor 2 |
|---|---|---|
| Cold/Warm | 0.88 | −0.35 |
| Dark/Light | 0.87 | −0.12 |
| Negative/Positive | 0.83 | −0.30 |
| Like/Unlike | −0.30 | 0.86 |
| Pleasant/Unpleasant | −.03 | 0.77 |
| Familiar/Unfamiliar | −0.44 | 0.53 |
| Simple/Complex | −0.39 | 0.45 |
| Sweet/Bitter | −0.03 | 0.20 |

As shown in Table 3, odor had a significant main effect on both outcome measures. For Affective Dimensionality, a significant effect of odor was observed, $F(1, 47.15) = 76.864$, p < .001, with a large effect size (partial $\eta^2 = 0.62$). Similarly, for Perceptual Valence and Complexity, the effect of odor was also significant, $F(1, 47.78) = 85.477$, p < .001, with a large effect size (partial $\eta^2 = 0.64$).

Table 4 lists the EMMs derived from the mixed effects model. For Affective Dimensionality, ratings were higher for paintings produced under the strawberry odor condition (EMM = 11.33, standard error [SE] = 0.546, 95% confidence interval [CI] [10.25, 12.42]) than for those produced under the rose odor condition (EMM = 8.77, SE = 0.546, 95% CI [7.69, 9.86]). In contrast, for Perceptual Valence and Complexity, higher ratings were observed for the rose condition (EMM = 12.2, SE = 0.169, 95% CI [11.9, 12.5]) than for the strawberry condition (EMM = 10.4, SE = 0.169, 95% CI [10.0, 10.7]).

Planned pairwise comparisons between odor conditions were conducted using the EMMs with Bonferroni correction (S2 Table, pairwise contrasts). For Affective Dimensionality, the contrast between rose and strawberry was significant (rose-strawberry = 2.56, SE = 0.295, $t(48.2) = 8.666$, p < .001), corresponding to a large effect size (Cohen's $d = 1.12$, 95% CI [0.865, 1.38]). For Perceptual Valence and Complexity, the contrast was also significant but in the opposite direction (rose-strawberry = −1.81, SE = 0.199, t(49.1) = −9.095, p < .001), again indicating a large effect size (Cohen's $d = −1.06$, 95% CI [−1.30, −0.829]).

In the third rating phase, the raters predominantly chose yellow and orange for the strawberry odor, while opting for more green and blue tones for the rose odor (Figs 4 and 5).

EFA was conducted on the semantic differential ratings to examine the underlying structure of the odor evaluations. Sampling adequacy was supported (KMO = 0.82), and Bartlett's test of sphericity indicated that the correlation matrix

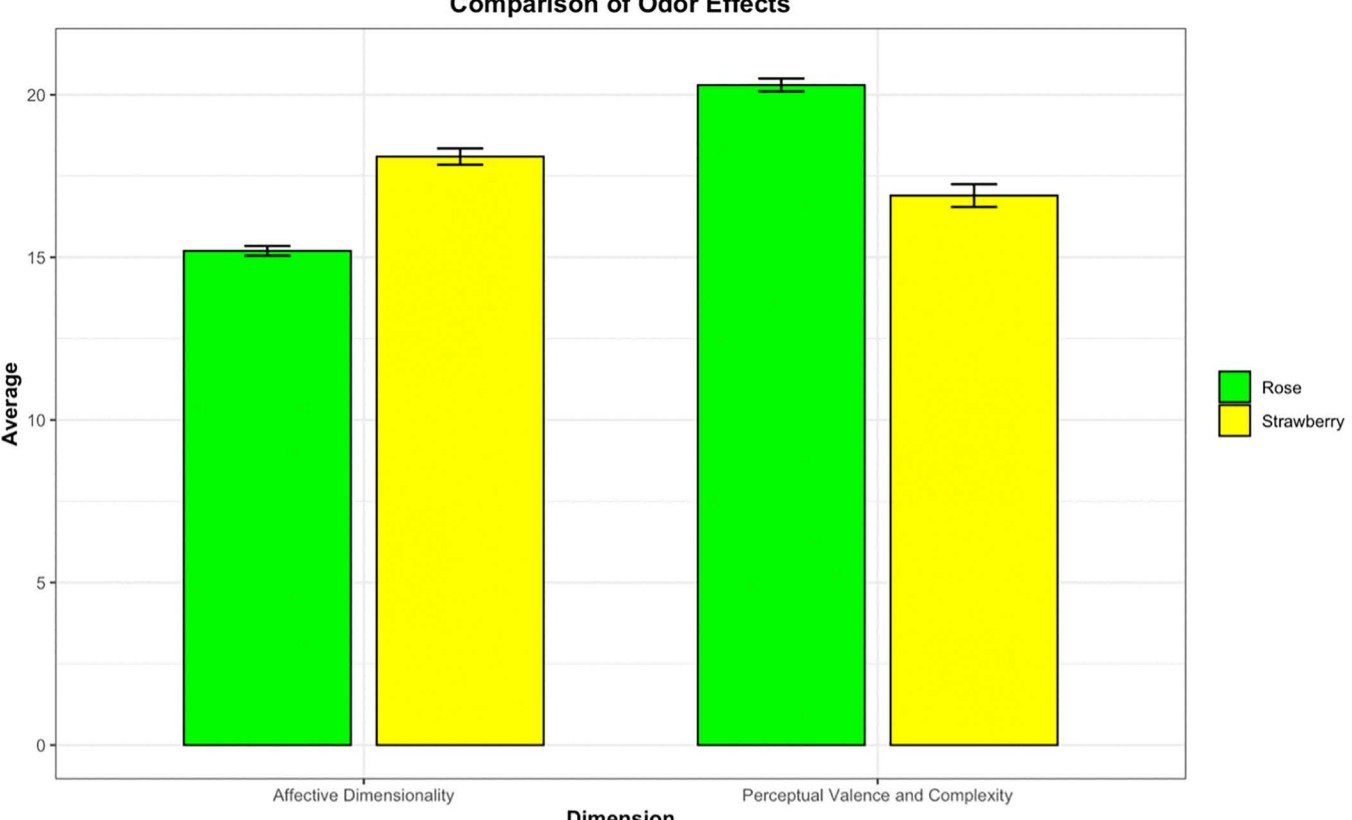

**Fig 3. The averages of factor scores in the evaluation phase.**

**Table 3. Mixed-effects model results for odor effects in relation to Affective Dimensionality and Perceptual Valence and Complexity.**

| Outcome | Fixed effect | F | df1 | df2 | p | partial_eta2 |
|---|---|---|---|---|---|---|
| Affective Dimensionality | Odor | 76.864 | 1 | 47.15 | < .001 | 0.62 |
| Perceptual Valence and Complexity | Odor | 85.477 | 1 | 47.78 | < .001 | 0.64 |

**Table 4. Estimated marginal means (EMMs) and pairwise comparisons.**

| Outcome | Odor | EMM | SE | df | CI_lower | CI_upper |
|---|---|---|---|---|---|---|
| Affective Dimensionality | strawberry | 11.33 | 0.546 | 79.5 | 10.25 | 12.42 |
| Affective Dimensionality | rose | 8.77 | 0.546 | 79.5 | 7.69 | 9.86 |
| Perceptual Valence and Complexity | strawberry | 10.4 | 0.169 | 84.3 | 10 | 10.7 |
| Perceptual Valence and Complexity | rose | 12.2 | 0.169 | 84.3 | 11.9 | 12.5 |

was suitable for factor analysis, $\chi^2(28) = 269.61$, $p < .001$. Although parallel analysis suggested a predominantly unidimensional structure, a two-factor solution was explored to facilitate the interpretation of the perceptual and evaluative components.

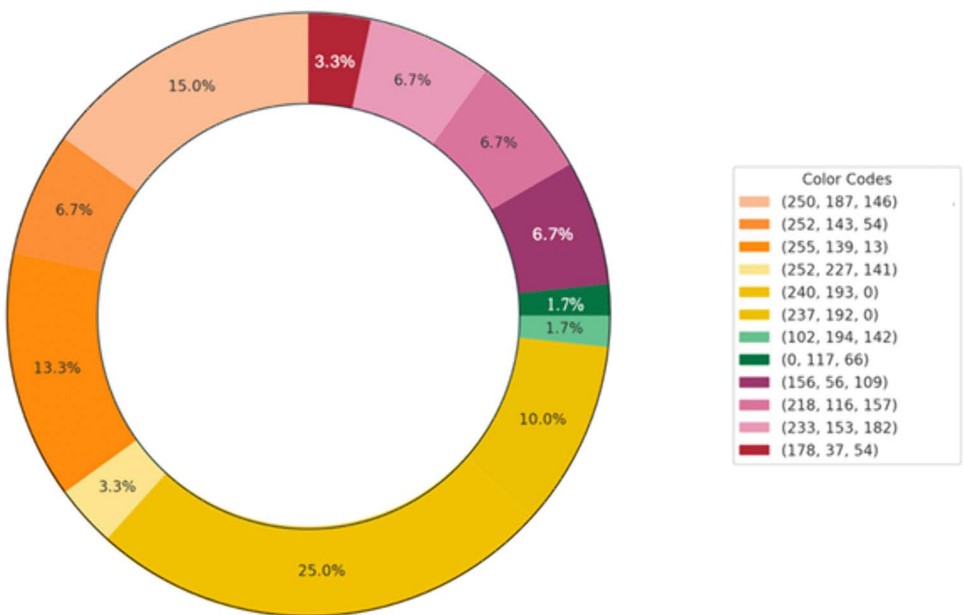

**Fig 4. Colors selected by participants for the strawberry odor.**

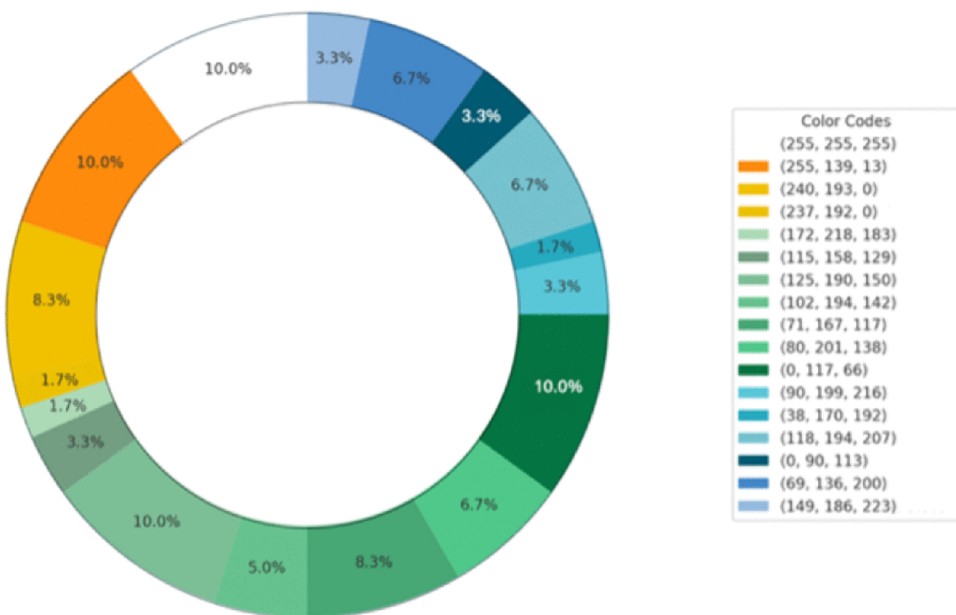

**Fig 5. Colors selected by participants for the rose odor.**

The rotated factor matrix revealed a meaningful two-factor structure (see Table 5). Factor 1 was primarily characterized by high loadings on *Simple-Complex* (0.82), *Sweet-Bitter* (0.63), and *Dark-Light* (−0.53), indicating an affective evaluative dimension reflecting qualitative and sensory attributes of odor perception. Factor 2 showed strong loadings for *Cold-Warm*

**Table 5. Varimax-rotated factor loadings for the principal axis factor analysis of odor ratings.**

| Variables | Factor 1 | Factor 2 |
|---|---|---|
| Sweet/Bitter | 0.63 | −0.20 |
| Simple/Complex | 0.82 | −0.21 |
| Dark/Light | −0.53 | 0.43 |
| Pleasant/Unpleasant | 0.43 | −0.53 |
| Familiar/Unfamiliar | 0.10 | −0.15 |
| Like/Unlike | 0.10 | −0.38 |
| Cold/Warm | −0.57 | 0.69 |
| Negative/Positive | −0.11 | 0.46 |

(0.69), *Pleasant-Unpleasant* (−0.53), *Negative-Positive* (0.46), and *Like-Unlike* (−0.38), representing a perceptual-sensory dimension associated with emotional valence and hedonic appraisal.

Several items, such as *Pleasant-Unpleasant* and *Dark-Light*, exhibited cross-loadings across both factors, suggesting a partial overlap between the perceptual and affective components of odor experience. Together, these two factors accounted for 41.1% of the total variance, supporting a distinction between perceptual-sensory attributes and affective evaluations, while acknowledging their interrelated nature in odor perception (Fig 6).

As illustrated in Fig 7 and S3 Table, the strawberry odor received significantly higher ratings than the rose odor in the perceptual-sensory dimension, whereas the rose odor was rated higher in the affective evaluative dimension. To examine whether the odor evaluations differed across perceptual dimensions, composite scores were computed for an affective evaluative dimension and a perceptual-sensory dimension based on the factor analysis results. A linear mixed-effects model was fitted with odor (rose vs. strawberry), dimension (affective evaluative vs. perceptual-sensory), and their interaction as fixed effects, and the participants as a random intercept. The analysis revealed a significant main effect of dimension, $F(1, 236) = 222.21$, $p < .001$, indicating overall higher ratings in the perceptual-sensory dimension. The main effect of odor was not significant, $F(1, 236) = 0.48$, $p = .49$. Importantly, a significant odor × dimension interaction was observed, $F(1, 236) = 29.06$, $p < .001$, indicating that the odor effects differed across the evaluative dimensions (Table 6). Post-hoc comparisons with Bonferroni correction showed that the rose odor received higher ratings than the strawberry odor in the affective evaluative dimension ($p = .001$), whereas the strawberry odor was rated higher than the rose odor in the perceptual-sensory dimension ($p < .001$).

Descriptions of the odors in the rating phase are presented in Table 7. For the strawberry odor, 41.67% of the participants labeled it as "strawberry," followed by 23.33% associating it with "fruit," 18.33% with "gummy," 15.00% with "candy," and 1.67% with "berry." For the rose odor, 45.00% of the participants identified it as "rose," while 40.00% labeled it as "flower," followed by smaller percentages associating it with "detergent" (6.67%), "shampoo" (5.00%), and "perfume" (3.33%). This outcome closely mirrors the results obtained during the painting phase.

## Discussion

This study examined the influence of olfactory stimuli on artistic expression and emotional perception. The results indicated that the participants associated distinct colors with each odor. Strawberry odor was associated with yellow and orange, whereas rose odor was associated with green and blue. Factor analysis of painting evaluations revealed two main factors: 1) Affective Dimensionality and 2) Perceptual Valence and Complexity. The results revealed that paintings created under the strawberry odor were rated as lighter, warmer, and more positive. The identification of only two primary factors, Affective Dimensionality and Perceptual Valence and Complexity, differs from Osgood's classical semantic differential model, which posits three universal dimensions (Evaluation, Potency, and Activity) [23]. This deviation may stem from

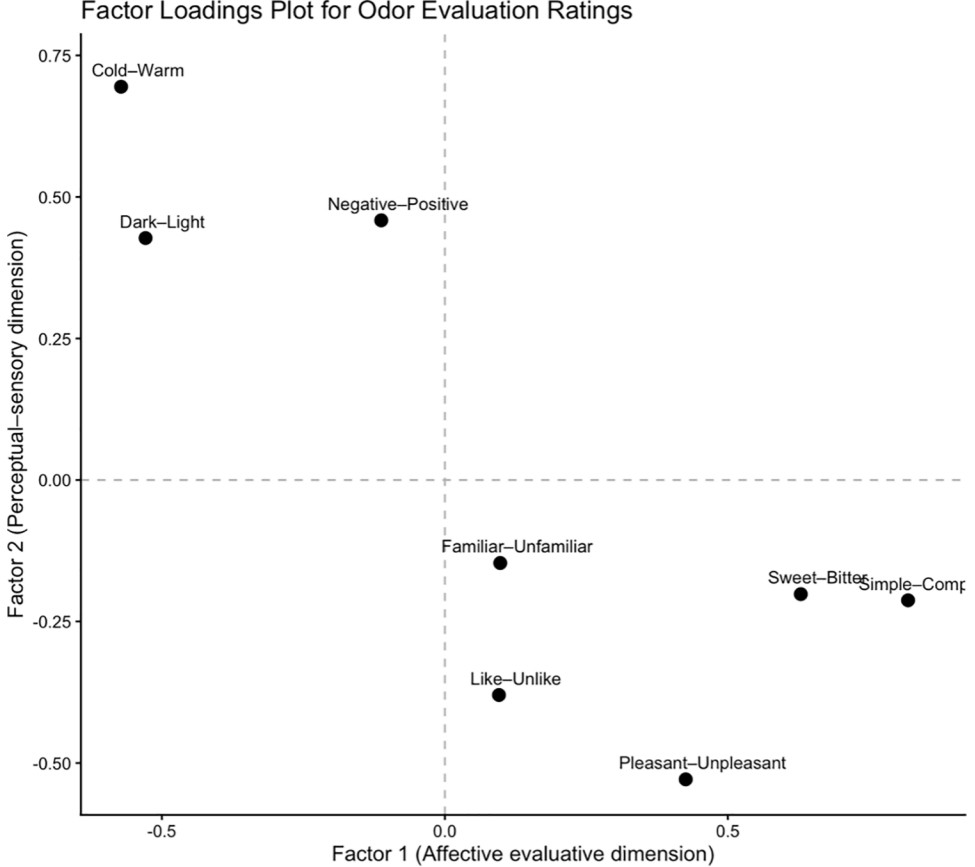

**Fig 6. Varimax-rotated factor loading plot for the odor-rating exploratory factor analysis (principal axis factoring).**

the specific nature of artistic and olfactory tasks, which likely emphasize affective and evaluative dimensions over those related to strength or intensity [13,25].

In the third phase, 60 additional participants (raters) assessed the odors using semantic differential words, in like manner to the evaluators in relation to the paintings. The results suggest that the paintings reflected odor impressions along the two primary dimensions. The rating phase revealed distinct patterns in how the raters perceived and labeled odors. The raters perceived the strawberry odor as warmer, more unfamiliar, and more positive than the rose odor. Except for complexity and sweetness, these perceptions aligned with the results of Phase 2 and the evaluators' assessments of the paintings. The strawberry odor was commonly associated with sweet and fruity characteristics, with the raters using labels such as "strawberry," "fruit," "gummy," and "candy." These associations reflect this odor's ability to evoke a broad spectrum of inter-pretations associated with sweet and familiar qualities. The raters selected a yellow color for its association with strawberry odor. Yellow is frequently associated with sweet flavors in various contexts. For example, Woods and Spence reported that yellow is associated with sweetness, although pink and red were more dominant in this regard [26]. The raters frequently described the rose odor using terms such as "rose" and "flower," while some associated it with various products such as "detergent," "shampoo," or "perfume," which often feature floral fragrances. These results suggest that the rose odor is perceived as distinct and strongly linked to its floral nature, with some participants interpreting it in the context of everyday products with odors. Supporting this speculation, the painters used more "green" for the rose odor and more "yellow" and warm colors for the strawberry odor in their paintings, which matched the colors they associated with each odor.

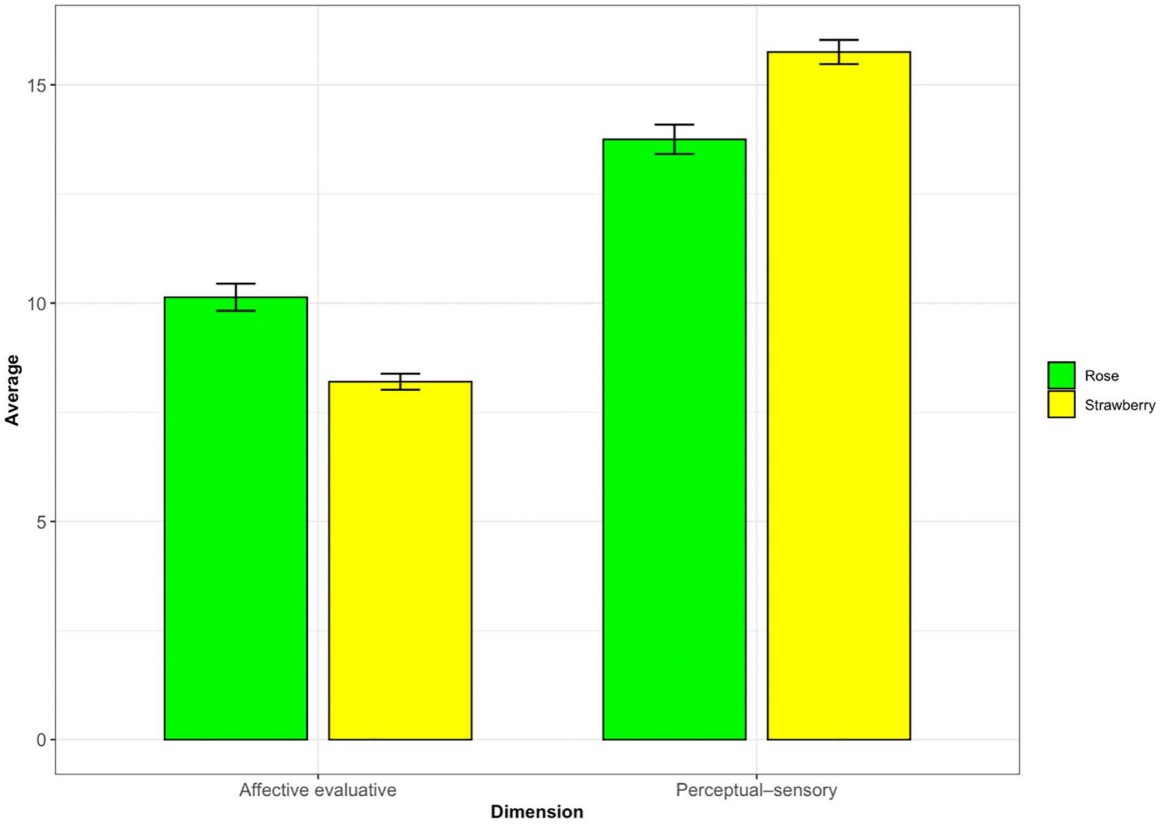

**Fig 7. Comparison of rose and strawberry odor ratings across factors.**

**Table 6. Linear mixed-effects model results for odor × dimension interaction.**

| Effect | Sum Sq | Mean Sq | Num df | Den df | F value | p value |
|---|---|---|---|---|---|---|
| Odor | 2.2 | 2.2 | 1 | 236 | 0.48 | 0.49 |
| Dimension | 1020.9 | 1020.9 | 1 | 236 | 222.2056 | < .001 |
| Odor × Dimension | 133.5 | 133.5 | 1 | 236 | 29.057 | < .001 |

These findings are consistent with the hypothesis that olfactory processing is associated with visual processing [12,27]. This study provides insight into the interplay between olfactory perception and artistic visual expression, highlighting the significance of this relationship in human creativity and sensory experience, and of the potential value of multisensory approaches to artistic expression. The odor-color interaction represents an important intersection of sensory modalities between olfaction and visual perception. Olfaction, with its profound impact on emotions and memories [28–30], demonstrates a unique capacity to enhance visual impressions in daily life. This intricate interplay between olfactory information and visual processes is exemplified by the phenomenon whereby certain odors enhance the attractiveness of visual images [31], creating a synergistic effect.

While Study 1 focused on the artistic and emotional impacts of odors, these findings raise broader questions about how olfactory stimuli guide people's choices and perceptual associations in other contexts. Study 2 aimed to address

**Table 7. Odor perception percentage distribution.**

| Word | Strawberry Odor (%) | Rose Odor (%) |
| --- | --- | --- |
| candy | 15.00 | |
| detergent | | 6.67 |
| flower | | 40.00 |
| fruit | 20.00 | |
| gummy | 18.33 | |
| rose | | 45.00 |
| shampoo | | 5.00 |
| strawberry | 41.67 | |
| perfume | | 3.33 |
| berry | 1.67 | |

these questions by investigating the role of these odors in object selection, color perception, and artistic output in a more controlled setting.

## Study 2

This study aimed to explore the impact of olfactory stimuli on object selection, color perception, and artistic expression. Specifically, it examined whether distinct odors, that is, strawberry and rose odors, influence the participants' choices of objects and associated color perceptions, and how these odors shape artistic outputs and evaluative judgments. The researchers hypothesized that participants exposed to either the strawberry or the rose odor would show a preference for specific colors (warm colors such as yellow for the strawberry odor and green for the rose odor), aligning with the findings of Study 1. Additionally, the participants were expected to select conceptually related objects, such as strawberries for the strawberry odor and roses for the rose odor. Furthermore, even without prior knowledge of the presence of odors, the participants exposed to the odor condition were expected to demonstrate odor-influenced object selection and perceptual associations, unlike those in the control group.

### Materials and methods

**Participants.** Sixty participants (30 women and 30 men; mean age = 21.9, SD = 1.9), all from Tohoku University, were randomly assigned to one of three groups corresponding to specific odor conditions. Study 2 was conducted from April 1, 2024 to June 30, 2024. Written informed consent was obtained from all participants, and the study was approved by the Ethical Committee for Medical Research on Humans at the Kawauchi Campus, Tohoku University (Approval No. 2023−011).

**Materials.** This study used two distinct odors, strawberry and rose, consistent with those used in Study 1. Odor presentation was standardized using an Aroma Breeze Nova Diffuser (Patchoul, Japan). The diffuser was operated using an internal absorbent cotton pad to which the essential oil was applied, which served as the sole odor source throughout the experiment. In the experimental environment, five artificial plastic objects were strategically arranged to present diverse visual options: yellow lemons, white roses, red strawberries, green pumpkins, and black eggplants, all from a local supermarket. The participants were instructed to paint their perceptions using a 24-color acrylic paint set (Fantastory, Model FAN-BOT-60X24-XBZ-EU), which included a wide spectrum of primary, secondary, and neutral shades. Additionally, the participants were provided with 30 brushes of varying sizes and A4-sized canvases. Colors specifically associated with each odor were identified using a Nihon Shikiken panel. The paintings were photographed using a digital camera under controlled indoor lighting conditions, with camera parameters (ISO, aperture, and white balance), camera-painting distance, and background remaining constant across sessions.

**Procedure and design.** In Study 2, to promote homogeneous odor distribution, two identical Aroma Breeze Nova diffusers were positioned symmetrically at the two corners of the experimental room. In each session, 10 drops (approximately 0.5 mL) of the undiluted essential oil were applied to the internal absorbent cotton pad of each diffuser. All devices were activated 10 min prior to participant entry to allow stable odor dispersion throughout the room, and remained active during the entire experimental session to maintain continuous ambient exposure. The diffusers were positioned outside the participants' direct line of sight to minimize visual awareness of the odor source. The participants were not instructed to inhale directly from any device. The experimental conditions, including the diffusion parameters, room dimensions, and exposure time, were held constant across participants for each odor condition to ensure uniform stimulus delivery. The participants were randomly divided into three groups and assigned to one of three rooms: one filled with strawberry odor, one with rose odor, and one with no odor. Five objects were placed on a table. The participants were not informed of the odors when the experiment started. They were asked to choose one of the five objects placed on a table and paint it within 15 minutes. After the painting session, the participants were asked to report whether they noticed any odor and the colors they perceived from the odor in the color panel. Fig 8 illustrates the experimental setup. The image preprocessing and color extraction procedures were identical to those used in Study 1. All paintings were photographed under controlled indoor lighting conditions using fixed camera parameters (ISO, aperture, and white balance), camera-painting distance, and background, consistent with Study 1.

**Analysis.** The analysis was conducted using R software, and statistical evaluation and visual representation tools were applied. The data were processed to calculate the percentage distributions of object choices and odor identification across the experimental conditions. Visualizations such as donut charts were created to depict object choices and color associations under different odor conditions. Python was used for image processing, and the PIL was used to resize the paintings and transform them into pixel matrices. In addition, k-means clustering was applied to identify the 25 most prominent colors. Image preprocessing and dominant color extraction procedures, including standardized RGB conversion and spatial normalization, were identical to those used in Study 1.

**Results.** As shown in Figs 9, 10, and 11, the analysis of painting colors revealed notable differences across the three conditions. The participants in the strawberry-odorized room predominantly selected objects conceptually related to strawberries and lemons, incorporating yellow and pink tones into their paintings. Conversely, the participants in the rose-odorized room tended to choose white roses and used lighter colors, such as white and gray, in their artwork. In the control group, the participants showed greater diversity in object choice and used a wider range of colors in their paintings.

Table 8 shows the distribution of object choices across the odor conditions. A chi-square test of independence using Monte Carlo simulation revealed a significant association between odor condition and choice ($\chi^2 = 52.37$, $p < .001$), with a large effect size (Cramér's $V = 0.62$). Follow-up pairwise comparisons (Bonferroni-corrected) indicated that all the odor conditions differed significantly from one another.

Fig 12 and Fig 13 illustrate the participants' color selections for the odors. For the strawberry odor, the participants predominantly selected yellow, reflecting a preference for warm hues. Conversely, under the rose odor condition, green emerged as the most frequently chosen color, indicating a preference for cooler tones associated with this odor.

Overall, 50% of the participants identified the strawberry odor as "strawberry," 40% as "fruit," and 10% as "candy," whereas 80% of the participants correctly identified the rose odor as "rose," while 20% associated it with "flower" (Table 9). In the no odor condition, the participants reported not sensing any smell; these responses are labeled as "No" in Table 9.

For the reported odor names, the statistical analysis was restricted to the strawberry and rose odor conditions because the participants in the no-odor condition did not report odor labels. A chi-square test of independence using Monte Carlo simulation revealed a significant association between odor condition and reported odor name ($\chi^2 = 40.00$, $p < .001$), with a very large effect size (Cramér's $V = 0.96$), indicating distinct odor identification patterns across the conditions.

 

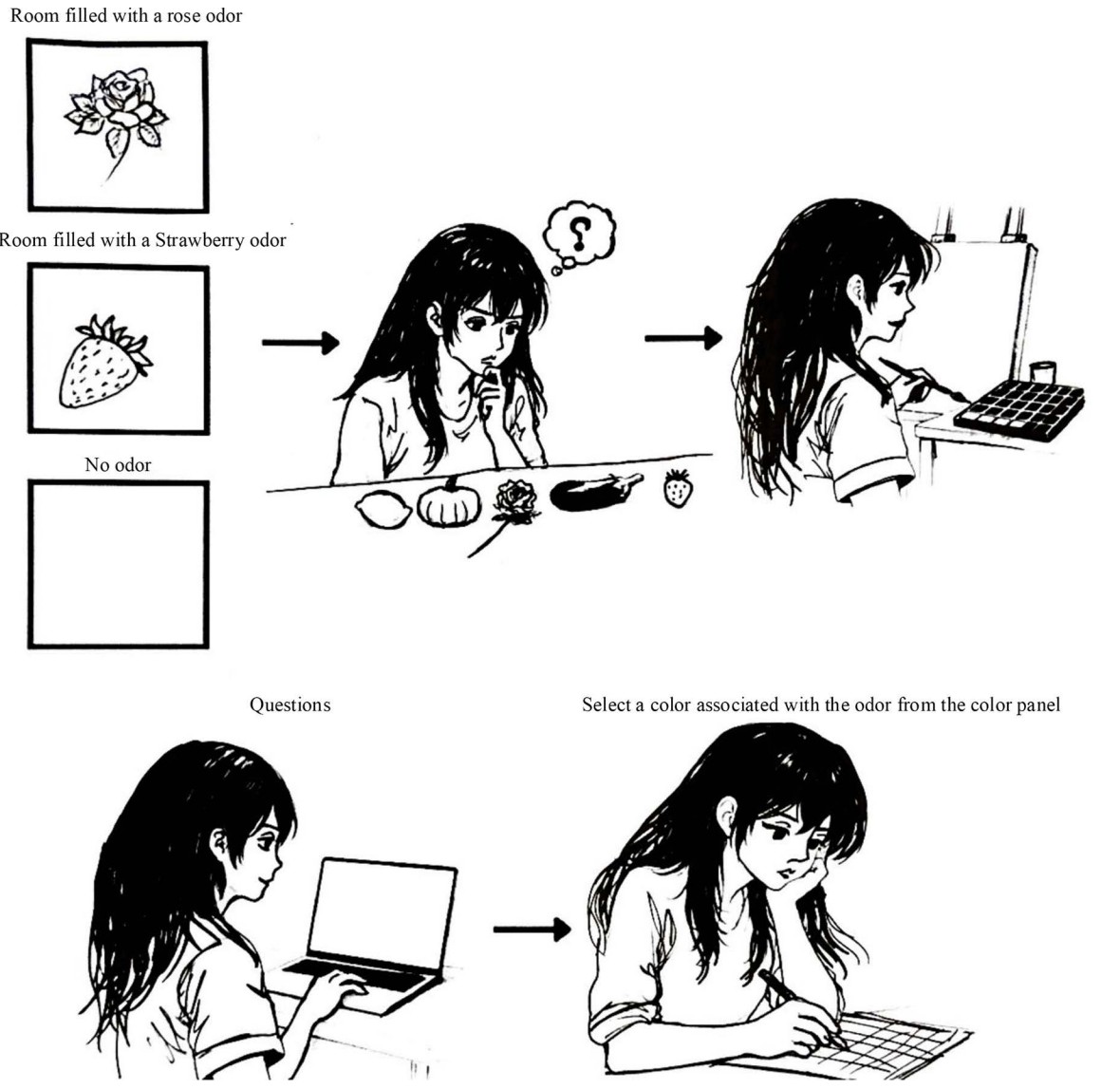

Room filled with a rose odor

Room filled with a Strawberry odor

No odor

Questions

Select a color associated with the odor from the color panel

**Fig 8. Diagram of the experimental setup.**

## Discussion

These findings suggest that olfactory stimuli are associated with differences in participants' object choices, color perceptions, and artistic outputs. The participants who were exposed to the strawberry odor predominantly selected objects conceptually or visually aligned with that odor, such as strawberries, whereas those in the rose odor condition gravitated toward the white rose. In contrast, the participants in the control group exhibited more diverse object choices.

While Study 1 primarily focused on the emotional and artistic impacts of the odors, Study 2 expanded the investigation by incorporating an object-selection component. Study 1 demonstrated that specific odors influenced the colors that the participants used in their paintings and the emotional impressions conveyed in the artworks, with warm colors such as yellow and pink associated with the strawberry odor and cooler tones such as blue and green associated with the rose odor.

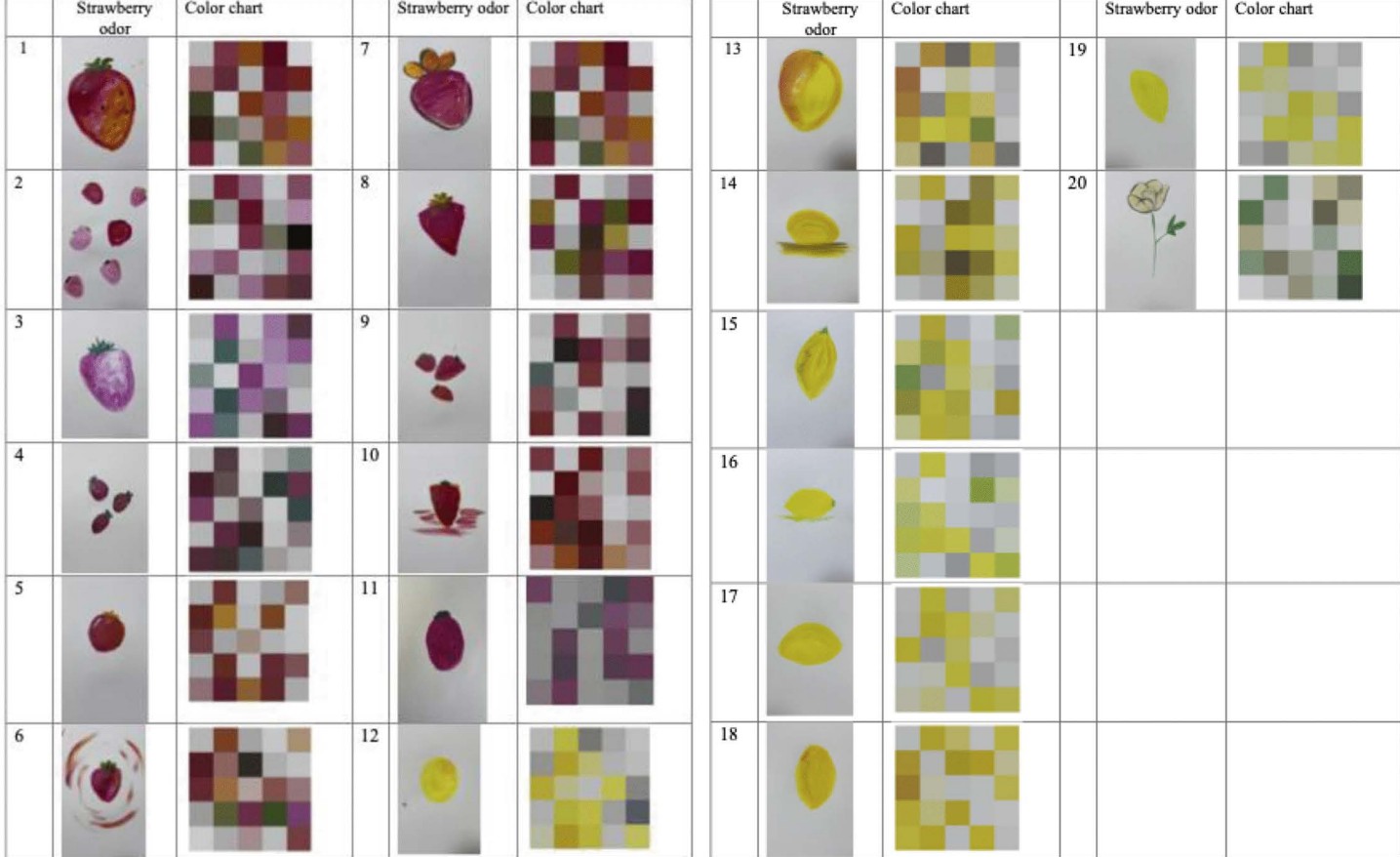

**Fig 9. Paintings in the strawberry-odorized room.**

Notably, in Study 1, the participants exposed to the rose odor used more green colors in their paintings, whereas in Study 2, they predominantly associated the rose odor with a white color and painted a white rose. This shift may be attributed to the presence of a white rose on the table, suggesting that visual perception and object availability played a role in shaping the participants' artistic choices. This difference highlights the interaction between olfactory and visual stimuli, indicating that, while odors influence artistic expression, available visual references can modulate these effects [15,32,33]. These results suggest that olfactory stimuli may extend beyond the enhancement of artistic creativity and contribute to the shaping of sensory-driven decisions. The emotional and psychological effects of warm colors appear to contribute to their association with sweetness. Warm colors are generally perceived as stimulating and exciting, which can enhance the overall eating experience and increase the likelihood of indulgent consumption [34].

## General discussion

This study explored the interplay between olfactory and visual perception in two studies. The findings provide converging evidence that odors may influence creative expressions and choice preferences. This interpretation is consistent with a substantial body of research on olfactory-visual and cross-modal interactions showing that sensory information from one modality can systematically shape perception, evaluation, and expressive outcomes in another modality, rather than serving merely as a contextual background [1,4,15]. Previous studies have shown that odors can

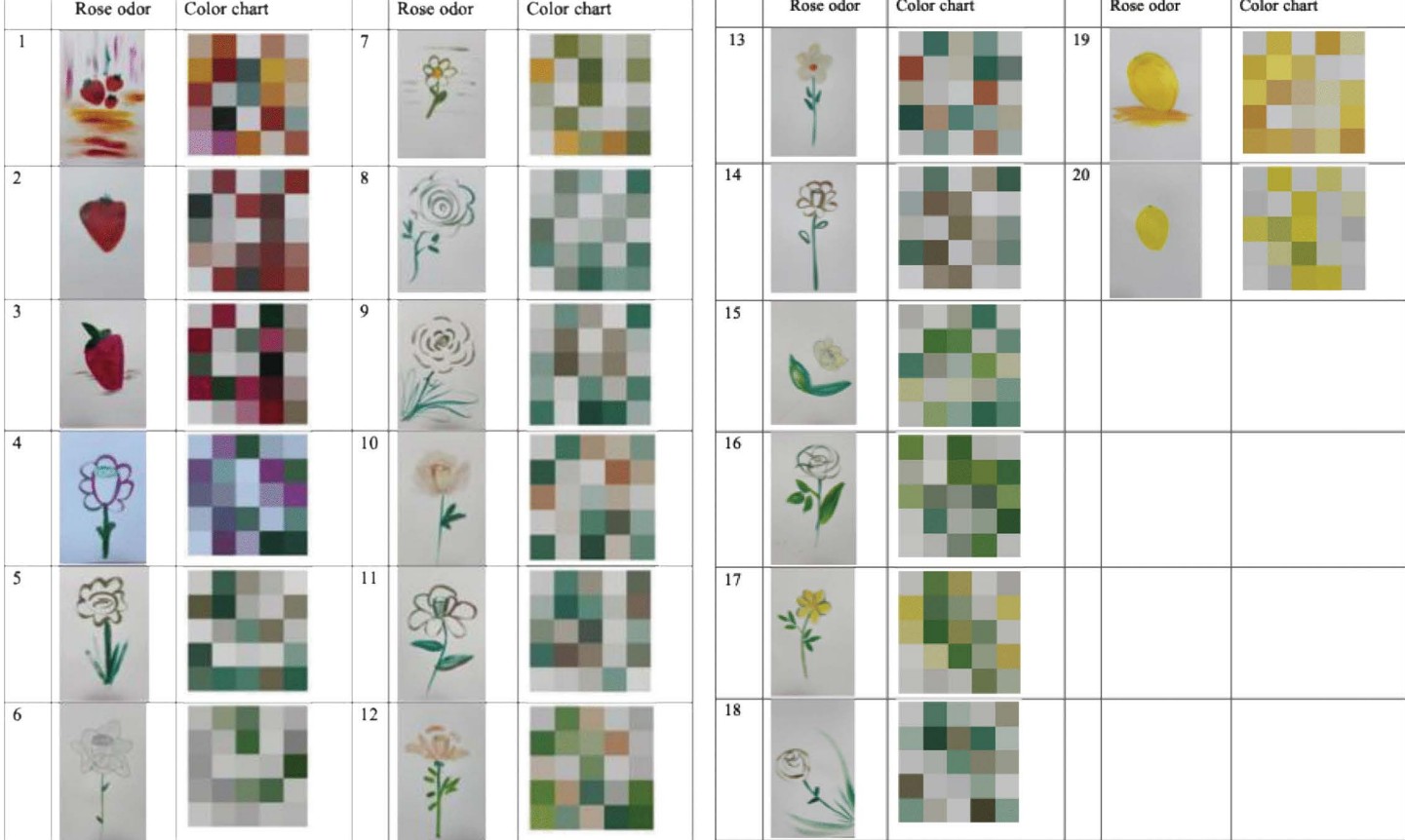

**Fig 10. Paintings in the rose-odorized room.**

bias visual attention and perceptual processing [4,35], alter aesthetic and evaluative judgments of visual stimuli, such as faces and artworks [32], and modulate emotional responses to visual scenes [36]. More broadly, research on cross-modal perception and synesthetic-like correspondences indicates that interactions between sensory modalities can influence higher-level cognitive processes, including preference formation, creative expression, and decision making [37–40]. This study provides further supporting evidence indicating that olfaction contributes to multisensory process-ing by shaping how visual information is perceived, emotionally interpreted, and translated into artistic and expressive decisions.

The results of Study 1 underscore the strong connection between odors and visual artistic output. The painters exposed to strawberry and rose odors created paintings with consistent color differences: warm colors dominated paintings inspired by the strawberry odor, while cooler colors were more prominent in paintings inspired by the rose odor. The evaluators' impressions of the paintings paralleled their evaluations of the corresponding odors. Across all the participants, artworks created under the influence of the strawberry odor were generally perceived as warmer, more positive, and brighter than those associated with the rose odor. Because mental imagery and semantic activation are known to guide visual expres-sion and aesthetic judgment, these differences provide a principled basis to expect systematic variation in color palette selection and in evaluations of paintings inspired by strawberry versus rose odors, while still allowing for partial overlap due to shared pleasantness and culturally learned color conventions.

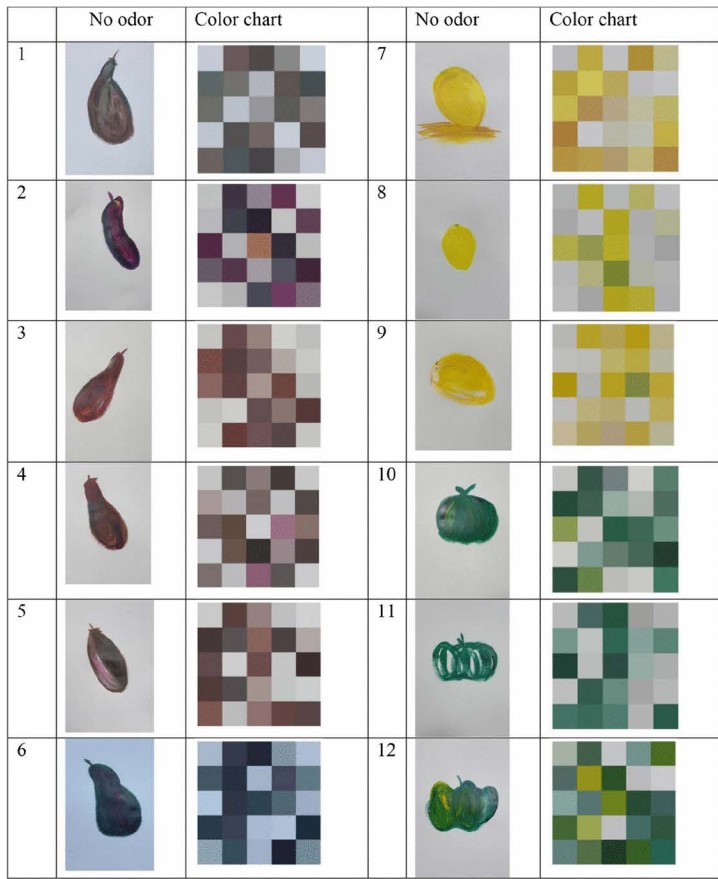

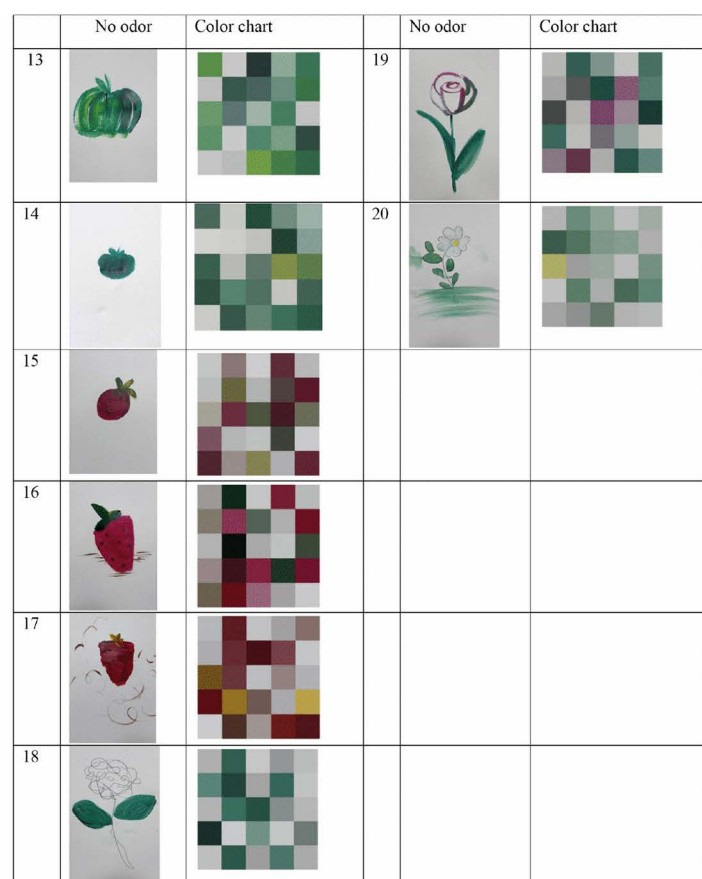

**Fig 11. Paintings in the control room with no odor.**

**Table 8. Object-selection frequency by condition.**

| Condition | Purple eggplant | Green pumpkin | Yellow lemon | White rose | Red strawberry |
|---|---|---|---|---|---|
| **Strawberry Odor** | 0 | 0 | 9 | 1 | 10 |
| **Rose Odor** | 0 | 0 | 2 | 15 | 3 |
| **No odor** | 6 | 5 | 3 | 3 | 3 |

Interestingly, these emotional qualities were also reflected in the participants' evaluations of the odors themselves. This consistency suggests a strong link between olfactory perception and visual-emotional expressions, indicating that odors can shape not only how we feel but also how we visually represent and interpret sensory experiences.

Study 2 expanded the scope of the investigation by examining how the odors influenced the participants' choices, providing evidence indicating that specific odors were associated with differences in object selection in terms of being conceptually or visually aligned with the odors. The participants in the odorized environments showed clear preferences for objects that matched the sensory qualities of the odor, whereas the control group displayed no distinct patterns. This suggests that certain odors can unconsciously guide preferences and decisions, likely through interactions with emotional and cognitive processes [41–43]. These results resonate with a substantial body of consumer research showing that ambient

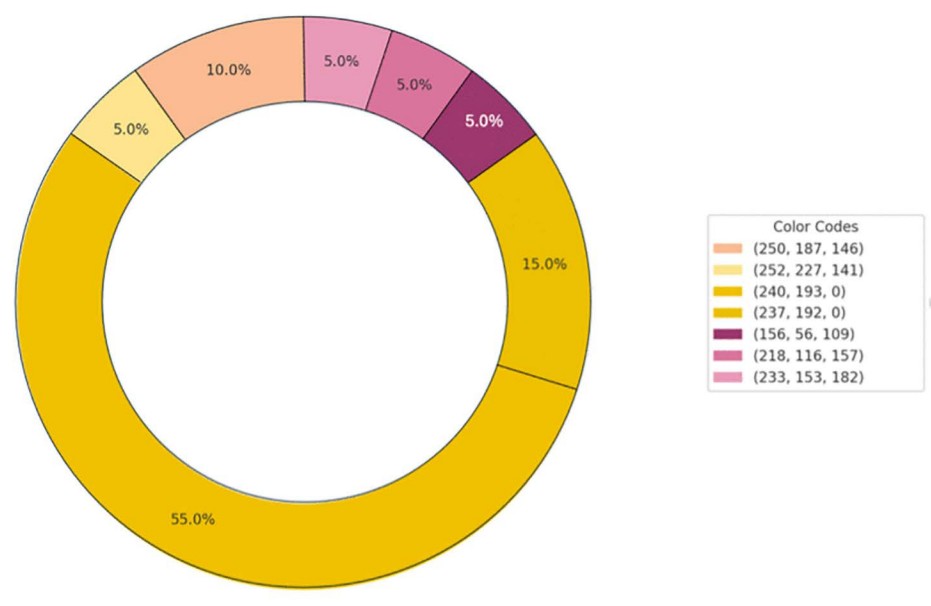

**Fig 12. Color associations with the strawberry odor.**

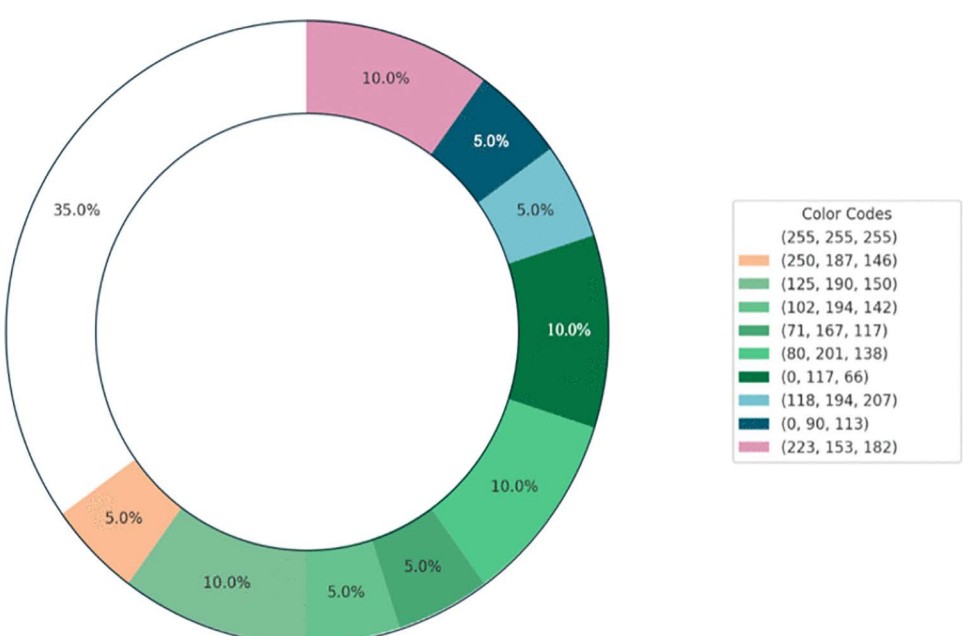

**Fig 13. Color associations with the rose odor.**

olfactory cues can influence attention and evaluations/preferences and can shift approach-related behaviors (e.g., time spent, impulse buying, and revisit intentions) in both controlled and real-world retail settings [41–47]. The participants predominantly chose warm colors such as yellow for the strawberry odor and cooler colors such as green for the rose odor

**Table 9. Odor identification results in Study 2.**

| Word | Strawberry Odor (%) | Rose Odor (%) | No odor (%) |
|---|---|---|---|
| Candy | 10 | | |
| Detergent | | | |
| Flower | | 20 | |
| Fruit | 40 | | |
| Gummy | | | |
| Rose | | 80 | |
| Shampoo | | | |
| Strawberry | 50 | | |
| No | | | 100 |

from the color panel, as in Study 1. Both studies showed similar and consistent labeling patterns for the odors across the different experimental phases. The participants commonly identified the strawberry odor with sweet and fruity labels, such as "strawberry," "fruit," "gummy," and "candy," while the rose odor was mainly associated with floral and fragrant labels such as "rose," "flower," "detergent," "shampoo," and "perfume." These consistent patterns across the naming, evaluation, and painting tasks indicate that certain odors evoke reliable, culturally, or personally grounded associations. The implications of these findings extend to various fields, including marketing and product design, where understanding the relationship between color and olfactory perception can inform strategies for enhancing consumer appeal. For instance, using warm colors in food packaging or presentations may potentially increase the perceived sweetness and desirability of products, thereby influencing consumer behavior [48,49].

The first main goal of this study was to investigate whether individuals perceive similar color ranges for the same odor when expressing their perceptions through paintings. The results of both studies support this idea, as the painters exposed to strawberry or rose odors consistently produced paintings with distinct and predictable color patterns. This suggests that these odors evoked shared sensory and emotional experiences, which were then translated into visual representations. Additionally, the evaluators perceived these paintings in ways that aligned with their odor perceptions, reinforcing the idea that olfactory stimuli trigger consistent cross-modal associations [50,51]. The second main goal was to explore how odors influence people's choices. Study 2 directly addressed this question by demonstrating that the participants in odorized environments displayed clear preferences for objects that were conceptually or visually aligned with the presented odor. This effect was absent in the control condition, highlighting that olfactory stimuli subtly but significantly shaped decision-making. The third main goal was to highlight the role of olfaction in color selection and evaluation. The consistent odor-labeling patterns observed across the participants suggest that certain odors evoke reliable, culturally, or personally grounded associations that extend beyond mere recognition to influence color perception and preference.

Together, these findings suggest that certain odors have a multifaceted influence on human perception and behavior. The effects of olfactory stimuli on visual and emotional processes highlight the interactions between sensory modalities in the brain and underscore the potential importance of odors in shaping creative expressions, decision-making, and visual and emotional perceptions. These insights have potentially far-reaching implications for fields such as art therapy, marketing, and environmental design, where the strategic use of odors may enhance creativity, user engagement, and well-being.

## Practical implications

This research has practical implications for various fields including art therapy, marketing, and environmental design. In creative and therapeutic contexts, certain odors are likely to evoke specific emotions or perceptions, facilitating artistic

expression and visual perception. In commercial settings, strategically deploying odors may influence consumer preferences and potentially guide decision making, enhancing the appeal of products or spaces.

### Limitations and future directions

This study has some limitations. First, the participants were limited to healthy young university students, which may affect the generalizability of the results. Second, controlled laboratory conditions may not fully reflect real-world environments, where odors interact with multiple sensory and contextual factors. Finally, this study focused on specific odors and stimuli, raising questions about how these effects may vary with different odors, cultural contexts, and individual differences in olfactory sensitivity.

Future research should explore these questions and investigate the long-term effects of repeated odor exposure, potential for habituation, and the neural mechanisms underlying cross-modal sensory interactions. Expanding this work to diverse populations and real-world settings could further illuminate the role of odors in influencing human perception, behavior, and creativity.

The study results provide converging evidence, suggesting a cross-modal correspondence. Strawberry and rose odors evoked warm (yellow/orange) versus cool (blue/green) color palettes, biasing the participants' color choices in their paintings and subtly influencing their object selection. These findings provide further evidence of the role of olfaction in multisensory perception and suggest that ambient odors may implicitly guide visual perception and creative decision-making.

## Supporting information

**S1 Table. Affective dimensionality and perceptual valence and complexity scores for each painting.**
(DOCX)

**S2 Table. Pairwise contrasts from estimated marginal means with Bonferroni correction.**
(DOCX)

**S3 Table. Sensory-affective dimensions and hedonic olfactory valence scores for each evaluator under rose and strawberry conditions.**
(DOCX)

## Acknowledgments

The manuscript was edited by Editage (Cactus Communications).

## Author contributions

**Conceptualization:** Zahra Davoudi.

**Data curation:** Zahra Davoudi.

**Formal analysis:** Zahra Davoudi, Nobuyuki Sakai.

**Funding acquisition:** Nobuyuki Sakai.

**Investigation:** Zahra Davoudi.

**Methodology:** Zahra Davoudi.

**Project administration:** Nobuyuki Sakai.

**Software:** Zahra Davoudi.

**Supervision:** Nobuyuki Sakai.

**Validation:** Nobuyuki Sakai.

**Writing – original draft:** Zahra Davoudi.

**Writing – review & editing:** Zahra Davoudi, Nobuyuki Sakai.

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
