## [Decision Letter · Decision Letter 0]

9 Dec 2025

Painting with odors: How olfactory stimuli influence artistic expression, emotional response, visual perception, and object selection.

PLOS One

Dear Dr. Sakai,

Thank you for submitting your manuscript to PLOS ONE. After careful consideration, we feel that it has merit but does not fully meet PLOS ONE’s publication criteria as it currently stands. Therefore, we invite you to submit a revised version of the manuscript that addresses the points raised during the review process.

In view of these reviews, I am asking for a revision in which you will have to respond to every point raised by the reviewers.

We look forward to receiving your revised manuscript.

Kind regards,

Bruno Alejandro Mesz, Ph.D.

Academic Editor

PLOS One

Journal Requirements:

[JST Grant Number JPMJPF2201 (NS)].

3. Please expand the acronym “JST” (as indicated in your financial disclosure) so that it states the name of your funders in full.

[This work was supported by JST Grant Number JPMJPF2201. This manuscript was edited by Editage (Cactus Communications K.K.).]

[JST Grant Number JPMJPF2201 (NS)]

5. We note that Figures 1a, 1c, and 7 in your submission contain copyrighted images. All PLOS content is published under the Creative Commons Attribution License (CC BY 4.0), which means that the manuscript, images, and Supporting Information files will be freely available online, and any third party is permitted to access, download, copy, distribute, and use these materials in any way, even commercially, with proper attribution. For more information, see our copyright guidelines: http://journals.plos.org/plosone/s/licenses-and-copyright.

1. You may seek permission from the original copyright holder of Figures 1a, 1c, and 7 to publish the content specifically under the CC BY 4.0 license.

6. Please upload a new copy of Figure 2 as the detail is not clear. Please follow the link for more information:  https://journals.plos.org/plosone/s/figures

Reviewers' comments:

Reviewer's Responses to Questions

**Comments to the Author**

1. Is the manuscript technically sound, and do the data support the conclusions?

Reviewer #1: Yes

Reviewer #2: Yes

2. Has the statistical analysis been performed appropriately and rigorously?

Reviewer #1: Yes

Reviewer #2: No

3. Have the authors made all data underlying the findings in their manuscript fully available?

Reviewer #1: Yes

Reviewer #2: Yes

4. Is the manuscript presented in an intelligible fashion and written in standard English?

Reviewer #1: Yes

Reviewer #2: Yes

Reviewer #1: No comments.

Reviewer #2: The manuscript presents two well designed experiments that explore an understudied but increasingly relevant research area. The experiments are creative, and generally methodological sound.

However, the manuscript would benefit from clarifications in methodology, restructuring for clarity, statistical reporting improvements and an thorough proofread for the English language.

1. Why strawberry and rose, what is your rationale for these two odours?

2. For you first reference it would be beneficial to add the following reference to help support the claim;

Ward, R. J., Ashraf, M., Wuerger, S., & Marshall, A. (2023). Odors modulate color appearance. Frontiers in psychology, 14, 1175703.

3. Line 51 – 54, would be worth mentioning there is more going on than just associative (i.e., hedonics), albeit associative is mostly like the strongest contributor in this case.

4. Investigation into the dynamics of visual attention reveals that background odors significantly affect visual behavior in tasks requiring rapid attentional deployment – reference?

5. It is hypothesized that exposure to strawberry and rose odors lead participants to

use distinct color palettes in their paintings and that paintings inspired by these odors are evaluated differently. – I think you need to strengthen this justification, one could argue that there could be a lot of overlap between these two colour pallets. i.e., green and red-pink, making them not distinct (maybe cultural impacted?). They are also both red, sweet, and pleasant smells.

6. You state that the odours were delivered with a diffuser. But at what concentration, was it consistent across participants?

7. Was there any ventilation in the room or attempt at odour contamination removal between participants?

8. Table 1, would be worth stating why Perfume and Berry are empty

9. What factor analysis algorithm did you use?

10. What criteria for factor retention did you use?

11. Table 2 might be better placed in the supplementary materials?

12. What factor rotation algorithm did you use?

13. Could you double check the GLM and AVOVA analysis the F-scores seem very high in places

14. Line 234 Bonferroni corrected? What were the specific comparisons and effect sizes, and exact p values? If your doing multiple tests where are the rest of the p-values?

15. Line 246 Principal component analysis?

16. For study 1, was the odour order presented randomly (i.e., first time it may or rose or strawberry or the control) or was it done consistently?

17. Can we see the first two loadings for the factor analysis in a graph?

18. Colour quantification more information is needed here, were images standardized for different lighting or scanned? Did the participants use identical paper and lighting setup (i.e., was the light controlled during the experiments). How was the number 25 choosen? Why k-means clustering, although a creative solution, were the colours converted to a perceptually modelled colour space or left as RGB?

19. Table 9, why is “No” now in the table, it doesn’t seem present in study 1, also what exactly does “No” mean in this context (no odour?)

20. Please be consistent when reporting you p-values (sometimes denoted as Sig), for example for Table 2, I assuming the * denote significance? What are their p values

21. I would go over the analysis again in its entirety, we need to know exactly what you did with the underlying data, we need to know all p-values

22. Inconsistent capitalisation in throughout the document

23. On lines 105-108 you mention a correlation might be present, could you perform this analysis or remove this statement if it isn’t applicable.

24. Table 8 can you perform statistical tests (possibly chi-squared tests) instead of just describing them?

25. Table 9 same issue above

26. Despite previous research illustrating the profound effects of odors on cognition and behavior [30–34], the results of this study expanded on existing findings by demonstrating both the nuanced and measurable impacts of olfactory stimuli on multisensory processing and artistic expression – please rephrase this sentence

27. This aligns with the findings from previous studies on olfactory and visual interactions[35,36]. They consistently showed that odors play a significant role in influencing artistic expression, guiding choices, and modulating emotional and visual perception. – I would expect more reference to be included in this statement, two seems very small given your claim here.

28. These results resonate with previous [37,38] studies on olfactory cues in consumer behavior, emphasizing the role of odors in shaping choices and attention. – again I would expect more references for this statement, a broad claim needs broad support.

29. Speaking of, you need to include more information about the odours themselves, how much mL of essential was used in the diffuser? How was the released controlled? Time? Intensity calibrated? Were the oils diluted or used neat? If diluted what was the ratio used, I assume this were mixed with water.

30. At the very end, can you place a very concise conclusion?

**Do you want your identity to be public for this peer review?** For information about this choice, including consent withdrawal, please see our For information about this choice, including consent withdrawal, please see our Privacy Policy .

Reviewer #1: No

Reviewer #2: **Yes:** Ryan J. WardRyan J. Ward

---

## [Author Response · Author response to Decision Letter 1]

16 Jan 2026

The manuscript presents two well designed experiments that explore an understudied but increasingly relevant research area. The experiments are creative, and generally methodological sound.

However, the manuscript would benefit from clarifications in methodology, restructuring for clarity, statistical reporting improvements and an thorough proofread for the English language.

Why strawberry and rose, what is your rationale for these two odours?

We thank the reviewer for raising this important point. We have now added a dedicated paragraph in the Introduction to clarify the rationale for selecting strawberry and rose odors. In brief, we chose these odors because (1) they are highly familiar and easily nameable in our international student population, (2) they are broadly matched in pleasantness and “sweet” hedonic tone, (3) they differ in semantic source category (edible fruit vs. non-edible flower/cosmetic product), and (4) there is existing evidence that fruity and floral odors show reliable odor–color correspondences. This combination allowed us to examine how odors with similar valence but different ecological and semantic properties are mapped onto colors and expressed in paintings.

Revised text in the manuscript (Introduction):

“Strawberry and rose odors were selected because they provide a theoretically controlled contrast along semantic and ecological dimensions while remaining comparable in hedonic valence. Both odors are highly familiar and easily identifiable across cultures, making them suitable for use with an international student population (R. S. Herz, 2004; Levitan et al., 2014). Although strawberry and rose odors are both typically perceived as pleasant and are often associated with red or pink hues, converging evidence indicates that odor–color associations are not determined by hedonic valence alone but are strongly influenced by semantic category and object-based knowledge(Deroy et al., 2013; Luisa Dematte, 2006). Strawberry represents a sweet, fruity, and edible odor that is closely linked to gustatory experience and food-related representations, whereas rose is a floral odor associated with botanical environments and culturally embedded symbolic meanings such as elegance or romance(Spence, 2020).”

For you first reference it would be beneficial to add the following reference to help support the claim;

Ward, R. J., Ashraf, M., Wuerger, S., & Marshall, A. (2023). Odors modulate color appearance. Frontiers in psychology, 14, 1175703.

We appreciate this suggestion. We have added Ward et al. (2023) as a reference in our Introduction to reinforce our statement about odors influencing color perception. The revised manuscript now includes Ward et al. (2023) as a supporting reference for the crossmodal influence of odors on color processing.

Revised text in the manuscript (Introduction):

The attributes of odor stimuli contribute to cross-modal correspondences between odors and visual characteristics, such as color dimensions(Ward et al., 2023).

Line 51 – 54, would be worth mentioning there is more going on than just associative (i.e., hedonics), albeit associative is mostly like the strongest contributor in this case.

Thank you for this helpful suggestion. We agree that odor–color correspondences cannot be explained solely by associative learning.

Accordingly, we revised Lines 72–88 to explicitly acknowledge the contribution of hedonic and affective factors alongside learned associations. The revised text now clarifies that while associative learning is likely the dominant mechanism, hedonic valence and emotional responses to odors also play a substantial role in shaping odor–color mappings.

Revised text in the manuscript (Introduction ):

“Odor–color correspondence should be interpreted within the broader framework of odor association learning rather than as an isolated crossmodal mapping. Evidence from taste–odor learning demonstrates that odors are preferentially associated with the emotional (hedonic) aspects of gustatory information, rather than with its qualitative sensory features. Using higher-order conditioning paradigms, Onuma and Sakai showed that odors acquire associative value primarily through emotional valence, with second-order conditioning emerging only when affective information is available, whereas associations based on sensory quality alone are comparatively weak(Onuma et al., 2016). Complementary human studies further indicate that visual information systematically modulates odor perception via top-down cognitive processes: visual cues such as colors or images evoke learned expectations that influence perceived odor intensity, preference, and identification(Sakai, 2005) These findings suggest that odor–visual correspondences, including odor–color associations, arise from learned multimodal associations grounded in hedonic evaluation and cognitive expectation. From this perspective, color functions as a visual manifestation of affective and semantic associations linked to odors, rather than as a direct sensory counterpart, providing a theoretical basis for understanding odor-induced modulation of visual expression. “

Investigation into the dynamics of visual attention reveals that background odors significantly affect visual behavior in tasks requiring rapid attentional deployment – reference?

Thank you for this comment. We have revised this statement and added appropriate references in the Introduction to explicitly support the claim that background odors influence visual attention. Specifically, prior studies have demonstrated that ambient odors modulate rapid visual attentional processes, including reflexive orienting, visual search, and attentional capture. These references have now been added to lines 47–72 of the revised manuscript.

Revised text in the manuscript (Introduction):

“Recent studies indicate that ambient olfactory cues can bias rapid visual attention. For example, in an involuntary “attentional capture” paradigm, Michael et al. (2003) found that background odors modulated the reflexive orienting to sudden luminance onsets(Michael et al., 2003). In a follow-up task, the same group (Michael et al., 2005) showed that a trigeminal (irritant) odor (allyl isothiocyanate) increased both the magnitude and duration of attentional capture, whereas a pleasant rose‐like odor (phenyl ethyl alcohol) effectively abolished the capture effect and generally slowed processing(Michael et al., 2005). Parallel findings emerge in visual search and scanning tasks: for instance, Seo et al. (2010) used eye-tracking to show that smelling a congruent odor caused observers to make more and longer fixations on matching objects than in an odorless condition(Seo et al., 2010), and Seigneuric et al. (2010) similarly found that congruent ambient scents led participants to locate odor-associated targets more quickly during free viewing(Seigneuric et al., 2010). Chen et al. (2013) extended this by demonstrating in dot-probe and search paradigms that a smell reflexively draws the attentional “spotlight” to a semantically matching image, facilitating detection even against competing distractors(Chen et al., 2013). Neurophysiological evidence also supports odor-driven attentional shifts: Zhang et al. (2021) recorded EEG during an Attention Network Test and found that an unpleasant odor elicited larger early N1/N2 components to alerting cues and faster behavioral alerting responses, suggesting heightened preparatory attention for negative odors(Zhang et al., 2021). Castellotti et al. (2025) reported that ambient fruit odors congruent with the search target improved visual search performance, yielding higher accuracy and faster responses for matching targets than for incongruent or no-odor trials(Castellotti et al., 2025). These behavioral and neurocognitive studies suggest that background odors can bias rapid visual attentional deployment—via affective congruency and arousal mechanisms—altering eye movements, reaction times, and early neural processing of visual cues.”

It is hypothesized that exposure to strawberry and rose odors lead participants to

use distinct color palettes in their paintings and that paintings inspired by these odors are evaluated differently. – I think you need to strengthen this justification, one could argue that there could be a lot of overlap between these two colour pallets. i.e., green and red-pink, making them not distinct (maybe cultural impacted?). They are also both red, sweet, and pleasant smells.

We understand the concern and have strengthened the justification for this hypothesis. In the revised Introduction, we now elaborate that although both strawberry and rose are pleasant and share some red-associated imagery, they also have distinct qualities that could lead to different artistic outcomes. Strawberry (a sweet, fruity odor) might evoke imagery of fruits, whereas rose (a floral odor) might evoke nature or botanical imagery. We acknowledge that some overlap in color choices could occur, but we argue that the different semantic and emotional associations of a fruit versus a flower scent provide a basis to expect differences in color palettes and art evaluations. This clarification has been added to make our rationale more convincing.

Revised text in the manuscript : (Introduction)

“Although strawberry and rose odors are both typically perceived as pleasant and are often associated with red or pink hues, converging evidence indicates that odor–color associations are not determined by hedonic valence alone but are strongly influenced by semantic category and object-based knowledge(Deroy et al., 2013; Luisa Dematte, 2006). Strawberry represents a sweet, fruity, and edible odor that is closely linked to gustatory experience and food-related representations, whereas rose is a floral odor associated with botanical environments and culturally embedded symbolic meanings such as elegance or romance(Spence, 2020)”

You state that the odours were delivered with a diffuser. But at what concentration, was it consistent across participants?

We thank the reviewer for requesting further clarification regarding the odor diffusion procedure. In the revised manuscript, we have described the method in a device-independent manner. The odor stimuli consisted of vaporized aroma (essential oil) applied to absorbent cotton and presented to participants at a fixed distance, allowing controlled olfactory stimulation without direct contact. For each experimental session, a fixed amount of essential oil (10 drops; approximately 0.5 mL) was applied to the cotton and used neat (undiluted). Odor delivery was controlled procedurally by maintaining identical preparation methods, presentation distance, room size, and exposure duration across all participants within each odor condition. A fixed pre-exposure period was implemented to allow the odor to stabilize before participant entry, and presentation continued throughout the task.

The exact airborne concentration of the odor was not chemically quantified. However, consistency of odor exposure was ensured by standardized procedures and supported behaviorally by the high agreement in odor identification and labeling across participants, indicating comparable perceived odor intensity within each condition.

Was there any ventilation in the room or attempt at odour contamination removal between participants?

Yes, we took measures to prevent odor carryover between participants. We have now described in the Methods that after each participant’s session, the room was ventilated (doors/windows opened and an interval allowed for air circulation) and the diffuser was turned off well in advance of the next session. To prevent odor carryover, sessions involving different odor conditions were separated by a minimum interval of 48 hours, and the experimenter confirmed that no residual odor remained before the next session. These steps ensured that the baseline odor condition was reset and that no residual scent from a previous session would affect the next participant

Revised text in the manuscript: (Procedure and design )

“A minimum interval of 48 hours was maintained between sessions to minimize potential carryover effects. Before the start of each subsequent session, the experimenter confirmed that no residual odor from the previous condition remained in the exprimental room. We opened windows and used an electric fan to dissipate any remaining odor, ensuring that each participant started in a neutral olfactory environment. The order of odor conditions in Study 1 was counterbalanced across participants. Each painter experienced the two odor conditions (strawberry, rose) in random order.”

Table 1, would be worth stating why Perfume and Berry are empty

Thank you for this suggestion. In Study 1, participants did not report perceiving or labeling any odors as Berry or Perfume. As a result, no responses were recorded in these categories, which led to the empty cells in Table 1. To avoid potential confusion, we have revised Table 1 by removing the Berry and Perfume categories from the Study 1 table, so that it now includes only odor labels that were actually reported by participants in that study.

What factor analysis algorithm did you use?

Exploratory factor analysis was conducted using principal axis factoring (PAF) with varimax rotation. This information has been added to the Methods (Analysis section) of the revised manuscript.

Revised text in the manuscript: (Analysis section)

“An exploratory factor analysis (EFA) was conducted on the semantic differential ratings to identify underlying perceptual dimensions. Prior to factor extraction, sampling adequacy was confirmed using the Kaiser–Meyer–Olkin (KMO) measure and Bartlett’s test of sphericity. Factors were extracted using principal axis factoring (PAF) with varimax rotation.”

What criteria for factor retention did you use?

Thank you for this important question. In the revised manuscript, we explicitly state that we used three complementary criteria for factor retention:

(1) Kaiser’s criterion (eigenvalues > 1.0),

(2) visual inspection of the scree plot, and

(3) parallel analysis.

All procedures converged on a two-component solution, which we now report in the Methods and Results sections.

Revised text in the manuscript (Results section):

“An exploratory factor analysis of the eight semantic rating scales revealed a two-factor solution. Based on eigenvalues greater than 1 (λ₁ = 4.32; λ₂ = 1.07), the two factors together explained 67.3% of the total variance…”

Table 2 might be better placed in the supplementary materials?

We agree that this table can be moved out of the main text. In the revision, we have relocated Table 2 to the Supplementary Materials. This keeps the flow of the main manuscript concise while still providing those data for readers who are interested.

What factor rotation algorithm did you use?

We thank the reviewer for requesting clarification. We applied varimax rotation to the extracted factors. This information has been added to the Methods (Analysis section) of the revised manuscript.

Revised text in the manuscript (Methods – Analysis section):

“Factors were extracted using principal axis factoring (PAF) with varimax rotation.”

Could you double check the GLM and AVOVA analysis the F-scores seem very high in places

Thank you for raising this point. The statistical analyses were carefully re-examined. The originally reported ANOVA results were descriptive and did not fully account for the non-independence arising from repeated ratings by evaluators and repeated paintings by painters. To address this concern, the primary analyses were re-conducted using linear mixed-effects models with Odor specified as a fixed effect and Painter and Evaluator specified as random intercepts.

Using this approach, the effect of odor remained highly significant for both outcome measures (Table 4). The observed F-values reflect large and reliable effects, as supported by the corresponding effect sizes (partial η² = 0.62 for Affective Dimensionality and partial η² = 0.64 for Perceptual Valence and Complexity). The manuscript has been revised to clarify that statistical inference is based on mixed-effects models, with ANOVAs reta

---

## [Editor Report · Decision Letter 1]

8 Feb 2026

Dear Dr. Sakai,

Thank you for submitting your manuscript to PLOS ONE. After careful consideration, we feel that it has merit but does not fully meet PLOS ONE’s publication criteria as it currently stands. Therefore, we invite you to submit a revised version of the manuscript that addresses the points raised during the review process.

Dear Dr. Sakai,

Thank you for submitting the revised version of your manuscript, PONE-D-25-40477_R1, entitled “Painting with odors: How olfactory stimuli influence artistic expression, emotional response, visual perception, and object selection.”

I have now completed an independent editorial evaluation of the revised manuscript, together with your detailed response to the reviewer’s comments.

I am pleased to inform you that the revision represents a substantial and effective improvement over the original submission. The major concerns raised during peer review—particularly regarding theoretical framing, methodological transparency, statistical analysis, and reporting clarity—have been addressed thoroughly and appropriately.

At this stage, the manuscript is scientifically sound and suitable for publication in PLOS ONE, pending resolution of a small number of editorial-level issues outlined below. These do not require further peer review and can be addressed in a final minor revision.

Please revise the manuscript to address the following minor issues:

Clarify odor delivery procedure for internal consistency

In the Methods section, references are made both to the use of a diffuser and to odor presentation via absorbent cotton. Please clarify whether these represent complementary steps within the same protocol or alternative delivery methods, and specify clearly:

where the stated volume (10 drops / ~0.5 mL) applies, the mode of exposure (room odorization vs. near-participant presentation), timing and duration of exposure.

This clarification will improve procedural transparency and reproducibility.

Specify the image color normalization procedure

While the color-analysis pipeline is now substantially improved, the description of “global color normalization” remains too general. Please briefly specify the method used (e.g., white balance correction, channel scaling, histogram normalization) or explicitly indicate where this procedure is documented in the shared analysis scripts.

Minor language and typographic polishing

Please perform a final proofreading pass to correct minor typographic errors, spacing issues, and occasional non-native phrasing (e.g., missing spaces, spelling errors). Also ensure consistency in capitalization, hyphenation (e.g., “no odor” vs. “no-odor”), and statistical notation throughout.

Tone calibration (optional but recommended)

A small number of statements in the Abstract and Discussion could be slightly softened to avoid over-assertive phrasing (e.g., replacing “confirm” with “provide evidence for”). This is optional but encouraged for stylistic balance.

We look forward to receiving your revised manuscript.

Kind regards,

Bruno Alejandro Mesz, Ph.D.

Academic Editor

PLOS One
---

## [Author Response · Author response to Decision Letter 2]

11 Mar 2026

We sincerely thank the Academic Editor for the careful review and for the helpful suggestions to improve methodological clarity and transparency. We have addressed each point and revised the manuscript accordingly. Please find these corrections in the manuscript.

---

## [Editor Report · Decision Letter 2]

13 Mar 2026

Painting with odors: How olfactory stimuli influence artistic expression, emotional response, visual perception, and object selection.

PONE-D-25-40477R2

Dear Dr. Sakai,

We’re pleased to inform you that your manuscript has been judged scientifically suitable for publication and will be formally accepted for publication once it meets all outstanding technical requirements.

Kind regards,

Bruno Alejandro Mesz, Ph.D.

Academic Editor

PLOS One
---

## [Editor Report · Acceptance letter]

PONE-D-25-40477R2

PLOS One

Dear Dr. Sakai,

I'm pleased to inform you that your manuscript has been deemed suitable for publication in PLOS One. Congratulations! Your manuscript is now being handed over to our production team.

Kind regards,

on behalf of

Dr. Bruno Alejandro Mesz

Academic Editor

PLOS One